# Myopericarditis and Pericardial Effusion as the Initial Presentation of Systemic Lupus Erythematosus in a Patient with Sickle Cell Trait: A Case Report

**DOI:** 10.3390/jcm14030920

**Published:** 2025-01-30

**Authors:** Marlon Rojas-Cadena, Felipe Rodríguez-Arcentales, Jose Narváez-Cajas, Marlon Arias-Intriago, Karen Morales Orbe, Juan S. Izquierdo-Condoy

**Affiliations:** 1Medical Science Faculty, Universidad Católica del Ecuador, Quito 170525, Ecuador; 2Department Section of Histology, Faculty of Medical Science, Universidad Central del Ecuador, Quito 170402, Ecuador; 3One Health Research Group, Universidad de las Américas, Quito 170521, Ecuador

**Keywords:** lupus erythematosus systemic, myocarditis, pericarditis, pericardial effusion, sickle cell trait

## Abstract

**Background:** Systemic lupus erythematosus (SLE) is a chronic autoimmune disease with rare but severe cardiac manifestations, including myocarditis and pericarditis. The coexistence of SLE with sickle cell trait (SCT), an inherited hemoglobinopathy prevalent among individuals of African descent, is exceptionally rare and presents significant diagnostic challenges due to overlapping clinical features. **Objective:** To describe the case of an Afro-Ecuadorian male with SLE and sickle cell trait who developed an uncommon presentation of myopericarditis and pericardial effusion. **Case report:** A 48-year-old African American male with no prior medical history presented with persistent fever, polyarticular arthralgias, and pleuritic chest pain. Investigations revealed sickle cell trait (SCT) and myopericarditis with pericardial effusion, marking the initial manifestation of SLE. Diagnostic delays occurred due to overlapping symptoms and a family history of sickle cell disease. Laboratory findings showed elevated hemoglobin S (<50%), positive ANA (1:1280, coarse speckled pattern), and anti-Smith/RNP antibodies, meeting EULAR/ACR 2019 criteria for SLE. Cardiac MRI confirmed myopericarditis. Treatment with pulse methylprednisolone, oral prednisone, and mycophenolate mofetil resulted in clinical improvement, with stable disease control on immunomodulatory therapy during follow-up. **Conclusions:** This case highlights the diagnostic complexity of SLE in patients with SCT, particularly when presenting with myopericarditis as the initial manifestation. It emphasizes the importance of a comprehensive diagnostic approach and timely initiation of immunosuppressive therapy to optimize clinical outcomes. This report broadens the understanding of overlapping syndromes involving SLE and SCT.

## 1. Introduction

Systemic lupus erythematosus (SLE) is a chronic autoimmune disease characterized by multisystem involvement and a wide spectrum of clinical manifestations. While it can occur at any age, SLE predominantly affects women, with a female-to-male ratio ranging from 1.2:1 to 15:1. For instance, a hospital-based study in Birmingham, UK, reported a prevalence of 49.6 per 100,000 in women compared to 3.6 per 100,000 in men [1,2].

Cardiac involvement in SLE is frequent and can affect multiple structures, including the pericardium, myocardium, valves, and coronary arteries [3]. Pericarditis, identified in approximately 25% of patients during the disease course, often presents with pleuritic chest pain relieved by sitting upright [4]. In contrast, *lupus myocarditis* is a rare yet potentially life-threatening manifestation, with a reported mortality rate of up to 20% [5]. Symptoms of lupus myocarditis vary widely, from mild presentations such as fever and chest pain to severe acute heart failure. Diagnosis relies on cardiac magnetic resonance imaging (MRI) as the gold standard, although echocardiography may reveal abnormalities when MRI is unavailable [5].

Sickle cell disease (SCD) is an autosomal recessive disorder caused by the homozygous inheritance of the *HBB* gene, leading to hemoglobin S (HbS) [6]. In contrast, individuals with heterozygous inheritance, known as sickle cell trait (SCT), are often asymptomatic [7]. However, under specific conditions such as hypoxia, dehydration, or acidosis, SCT can result in complications, including hematuria, rhabdomyolysis, venous thromboembolism, and sudden death during strenuous activity [6,8]. Sickle cell hemoglobinopathies predominantly affect populations of African descent, making SCT particularly relevant in Afro-descendant groups [9].

The coexistence of SLE and sickle cell hemoglobinopathies, such as SCT, is exceedingly rare. Despite their rarity, these conditions share overlapping clinical features—including joint pain, renal involvement, and acute chest syndrome—that complicate the diagnosis of lupus in patients with hemoglobinopathies [8].

We present the case of an Afro-Ecuadorian male with SLE and sickle cell trait who developed an uncommon presentation of myopericarditis. This case underscores the diagnostic challenges posed by the intersection of these conditions, given their rarity and overlapping symptomatology.

## 2. Case Presentation

A 48-year-old African American male with no significant past medical history but a notable family history of sickle cell disease in three sisters, including one with small vessel vasculitis, presented to the emergency department with a two-week history of fever, night sweats, myalgias, polyarticular arthralgias, and headache. Initial outpatient management, based on a presumptive diagnosis of an upper respiratory tract infection, included analgesics and antibiotics. However, symptoms persisted, accompanied by 5 kg weight loss and the onset of precordial–pleuritic chest pain, worsened in the supine position and were relieved in the sitting position (Figure 1).

On re-evaluation, physical examination revealed a grade 1 mid-diastolic murmur at the tricuspid area. Laboratory tests showed anemia (hemoglobin: 11.4 g/dL), leukopenia (4070/μL), and other abnormalities (Table 1). Echocardiography demonstrated a preserved ejection fraction (71%) and a 3.9 mm hyperechogenic pericardium (Figure 2). A chest CT revealed an 8 mm thickened pericardium with mild pericardial effusion (Figure 3), while an ECG displayed nonspecific changes. Based on these findings, the patient was admitted with a diagnosis of fever of unknown origin and suspected acute pericarditis.

During hospitalization, hemoglobin levels dropped to 11.4 g/dL, and leukocyte count decreased to 4070/μL. Protein electrophoresis revealed elevated hemoglobin S (<50%) (Figure 4), confirming the diagnosis of sickle cell trait. Suspecting a sickle cell crisis, hydroxyurea was initiated. However, after three weeks, fever (>38.5 °C) and nonspecific symptoms persisted, with new-onset polyarticular pain, edema, and inflammation in the left elbow. Further investigation, prompted by persistent pericardial effusion, included immunological studies. Antinuclear antibody (ANA) titers were 1:1280 (reference positive >1:80) with a coarse nuclear speckled pattern, and anti-Smith/RNP antibodies were >200. Anti-dsDNA antibodies were negative. Based on these findings and the 2019 EULAR/ACR criteria, the patient was diagnosed with systemic lupus erythematosus (SLE), meeting the criteria for leukopenia, strongly positive ANA, anti-Smith antibodies, pericarditis, myocarditis, fever, and joint involvement. Peripheral blood smear revealed poikilocytosis and a dual red blood cell population but no sickled cells. Additionally, the test was negative.

Cardiac magnetic resonance imaging (MRI) confirmed the presence of myopericarditis (Figure 5). Nonsteroidal anti-inflammatory drugs (ibuprofen), were initiated, leading to partial relief of chest pain. Concurrently, a biopsy of cervical adenopathy revealed nonspecific necrotizing suppurative lymphadenitis. The patient’s Systemic Lupus Erythematosus Disease Activity Index (SLEDAI) score was 19, indicating severe systemic disease activity.

To investigate potential infectious causes, a bronchoscopy with bronchoalveolar lavage (BAL) was performed to test for tuberculosis, bacteria, and fungi. Real-time PCR for tuberculosis was negative, while cultures identified *Penicillium chrysogenum*. This colonization was asymptomatic and likely incidental, discovered during a routine pre-immunosuppressive therapy evaluation. The patient received a one-week course of itraconazole (200 mg twice daily) as antifungal treatment, which resolved the colonization without complications.

Following completion of antifungal therapy, treatment for SLE was initiated based on recommendations from the rheumatology department. The patient received three intravenous pulses of methylprednisolone (250 mg daily for three days), followed by a taper to oral prednisone. Mycophenolate mofetil (MMF) was subsequently introduced to maintain disease control. Additionally, two doses of intravenous immunoglobulin (IVIG, 0.4 g/kg per dose) were administered to address immune-mediated processes. This comprehensive treatment regimen resulted in significant clinical improvement and stabilization of the patient’s condition.

Upon discharge, the patient was prescribed a maintenance regimen consisting of prednisone (30 mg daily for 15 days), MMF (500 mg twice daily for 90 days), and hydroxychloroquine (200 mg daily for 90 days). At monthly outpatient follow-ups, the patient demonstrated stable clinical improvement with no new disease flares. Long-term immunomodulatory therapy has maintained symptom control, emphasizing the importance of a multidisciplinary approach in managing complex autoimmune and hematological overlap syndromes.

## 3. Discussion

This case highlights the exceptional rarity of cardiac complications in a male patient with coexisting SLE and SCT, a combination rarely reported in the literature. While individuals of African descent are predisposed to both conditions, their concurrent occurrence is infrequent, and presentation in males is rare [6]. The involvement of cardiac complications further heightens the diagnostic challenges, providing valuable insights into this uncommon and complex clinical scenario.

Although SLE predominantly affects women, men with SLE often exhibit more severe disease manifestations, including higher rates of renal and cardiovascular involvement. This increased severity may be attributed to delayed diagnosis, as SLE is less frequently suspected in men, and there may be potential differences in hormonal and genetic factors that influence disease expression. Furthermore, men may face an elevated risk of treatment-related adverse effects, such as infections, when using immunosuppressive therapies. These risks necessitate careful monitoring and individualized treatment strategies [10]. Additionally, psychosocial challenges unique to men with SLE—such as the stigma associated with having a disease perceived as “female-predominant”—may negatively impact adherence to treatment. Addressing these gender-specific considerations is essential for improving diagnostic accuracy, optimizing treatment strategies, and enhancing long-term outcomes in this underrepresented population. This case not only broadens our understanding of the interplay between SLE and SCT but also provides valuable insights into managing SLE in men, a demographic that remains underrepresented in clinical studies [10].

Diagnosing SLE in patients with hemoglobinopathies, such as SCT, presents significant challenges due to overlapping multisystem manifestations that can delay recognition. SCT itself may exacerbate renal dysfunction, as individuals with sickle cell syndromes are prone to accelerated renal decline compared to healthy individuals [7,8]. Although no definitive causal relationship exists between SCT and SLE, in regions with a high prevalence of hemoglobinopathies, such as Esmeraldas, Ecuador, clinicians must maintain a high index of suspicion and carefully evaluate clinical signs and symptoms to ensure timely diagnosis.

Cardiac involvement in SLE is well documented, occurring in 5–10% of symptomatic patients, with up to 50% having subclinical cardiac involvement identified at autopsy [11]. The underlying mechanism involves immune complex deposition, leading to inflammation and the injury of various cardiac structures, including the pericardium, myocardium, endocardium, valves, and coronary arteries [6,12,13]. While pericarditis is the most common cardiac manifestation, lupus myocarditis remains a rare but potentially life-threatening complication. Previous studies, such as Thomas et al., report an 8:1 female-to-male ratio for lupus myocarditis, further highlighting the rarity of myocarditis in male patients with SLE [13].

In contrast, cardiac complications associated with SCT primarily stem from chronic anemia and vasculopathy, with mortality rates reported at 20–32%, largely due to progressive heart failure. Chronic anemia induces a high-output cardiac state, which, over time, leads to cardiac remodeling and eventual left ventricular failure. Pulmonary hypertension, a frequent complication of vasculopathy, further exacerbates cardiovascular risk. While cardiac complications in SCT are well documented, no direct association between SCT and myocarditis has been established [8].

Myocardial involvement, including myocarditis, presents a diagnostic challenge due to its wide spectrum of clinical manifestations. The most commonly presenting symptoms include chest pain (85–95%), fever (approximately 65%), and dyspnea (19–49%). Other features such as palpitations, syncope, and fatigue are also frequently reported. Distinguishing myocarditis from pericarditis is particularly challenging, as both conditions often coexist (myopericarditis) and share overlapping symptoms. Electrocardiogram (ECG) findings, such as ST segment elevation, are common in both conditions, though more widespread elevations are typically associated with pericarditis. Elevated troponin levels are more indicative of myocarditis or myopericarditis than isolated pericarditis [14]. Echocardiography findings consistent with myocarditis include increased myocardial wall thickness and abnormal echogenicity, which may reflect inflammation and edema. Importantly, early on in the course of myocarditis, left ventricular (LV) dimensions are usually normal, and left ventricular ejection fraction (LVEF) is preserved in approximately 75% of patients [15]. Cardiovascular magnetic resonance imaging remains the reference standard non-invasive modality for detecting myocardial inflammation and associated abnormalities [14]. The combination of clinical assessment, laboratory findings, and advanced imaging techniques is essential for the accurate diagnosis and management of myocarditis in patients with SLE and SCT.

Lupus myocarditis is challenging to diagnose due to its nonspecific presentation, which may include fever, dyspnea, chest pain, and palpitations [5,6,11,16]. Our patient presented with fever, pleuritic precordial pain, and a mid-diastolic murmur, consistent with the symptoms reported in prior studies. Notably, myocarditis was the debut manifestation of SLE in this patient, which aligns with findings by Thomas et al., where nearly 60% of patients presented with myocarditis as the initial symptom of SLE [13]. Given its potential progression to arrhythmias, conduction disorders, dilated cardiomyopathy, and heart failure, early recognition and intervention are critical to improving outcomes [11].

The diagnosis of lupus myocarditis requires a multimodal approach, combining clinical findings, biomarkers, and imaging. In this case, the patient demonstrated elevated ANA titers (1:1280, coarse speckled pattern) and positive anti-Smith/RNP antibodies, while anti-dsDNA antibodies and complement levels (C3 and C4) remained normal. Although anti-dsDNA antibodies are frequently positive in lupus myocarditis, their absence in this patient broadens the spectrum of diagnostic features. Elevated high-sensitivity troponin I (31.5 ng/L) served as a reliable marker of myocardial injury, consistent with findings from Thomas et al., where troponin I was elevated in 80% of cases and correlated with recent-onset heart failure [13].

Endomyocardial biopsy (EMB) remains the gold standard for diagnosing myocarditis; however, its invasive nature, risks, and limited availability often restrict its use to severe or life-threatening cases [13]. Noninvasive imaging modalities, particularly cardiac magnetic resonance imaging (CMRI), are now the preferred diagnostic tools. CMRI enables tissue characterization and detects critical changes, such as myocardial edema, necrosis, and fibrosis, which are hallmarks of lupus myocarditis [17,18,19]. In our patient, CMRI confirmed the diagnosis of myopericarditis and played a pivotal role in assessing the extent of myocardial involvement.

Managing lupus myocarditis is challenging due to the limited evidence on optimal treatment strategies. Current recommendations advocate the use of high-dose corticosteroids, often in combination with immunosuppressive therapies such as cyclophosphamide, MMF, or azathioprine [11,13]. In refractory cases, biologics like rituximab have demonstrated success, particularly in pediatric populations [13]. In this case, the administration of pulse methylprednisolone followed by oral prednisone and MMF resulted in significant clinical improvement, highlighting the critical role of early, multidisciplinary intervention and tailored immunosuppressive therapy.

The unique intersection of autoimmune and genetic conditions in this patient creates a complex clinical scenario characterized by overlapping and exacerbated complications. Hematologic challenges include severe anemia, autoimmune hemolytic anemia aggravated by SCT-related hemolysis, leukopenia, thrombocytopenia, and an increased thrombotic risk from antiphospholipid antibodies and SCT-induced venous thromboembolism. Renal complications are also significant, involving coexisting lupus nephritis and SCT-associated nephropathy, which present as proteinuria, hematuria, and progressive renal dysfunction. Vascular complications, such as microvascular ischemia, Raynaud’s phenomenon, and livedo reticularis, are worsened by SCT-induced sickling. Additionally, cardiopulmonary manifestations, including acute chest syndrome, pulmonary hypertension, and pericardial effusion, are exacerbated by hypoxia and vascular compromise. Neurologic complications, such as cerebrovascular events and cognitive dysfunction, further add to the complexity, alongside dermatologic findings like skin ulcers and impaired wound healing, which reflect the multifaceted challenges of managing this dual pathology [8].

This case expands the limited literature on lupus myocarditis in male patients with sickle cell trait, highlighting the diagnostic complexities posed by overlapping clinical features and the rarity of this association. It underscores the importance of a comprehensive diagnostic approach that integrates clinical evaluation, biomarker assessment (e.g., elevated troponin I), and advanced imaging tools such as CMRI. Early recognition and timely intervention with immunosuppressive therapy remain crucial for preventing complications and improving clinical outcomes in lupus myocarditis.

## 4. Conclusions

This case highlights an exceptional presentation of myocarditis as the initial manifestation of SLE in a male patient of African descent with sickle cell trait, a particularly rare combination given the predominance of SLE in women and the low reported prevalence of these conditions occurring together.

The coexistence of SLE and hemoglobinopathies, such as sickle cell trait, poses a diagnostic challenge due to overlapping multisystem symptoms, which can delay timely recognition and treatment. In regions with a high prevalence of hemoglobinopathies, such as Esmeraldas, Ecuador, physicians must maintain heightened diagnostic awareness and conduct thorough clinical evaluations to ensure the early identification and appropriate management of such rare presentations.

## Figures and Tables

**Figure 1 jcm-14-00920-f001:**
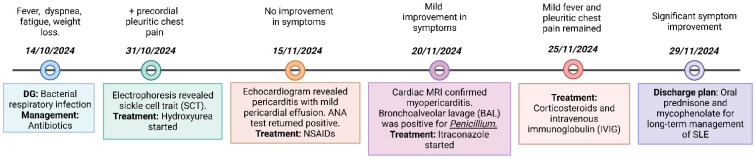
Timeline of patient.

**Figure 2 jcm-14-00920-f002:**
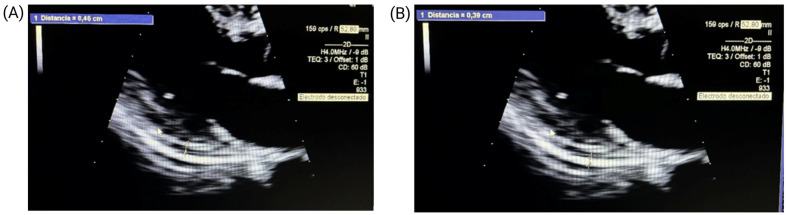
Echocardiographic findings. Left ventricular concentric remodeling with preserved global and segmental wall motion and systolic function (ejection fraction: 71%). (**A**) Observed 4 mm pericardial effusion in the posterior sac. (**B**) Hyperechogenic pericardium measuring 3.9 mm.

**Figure 3 jcm-14-00920-f003:**
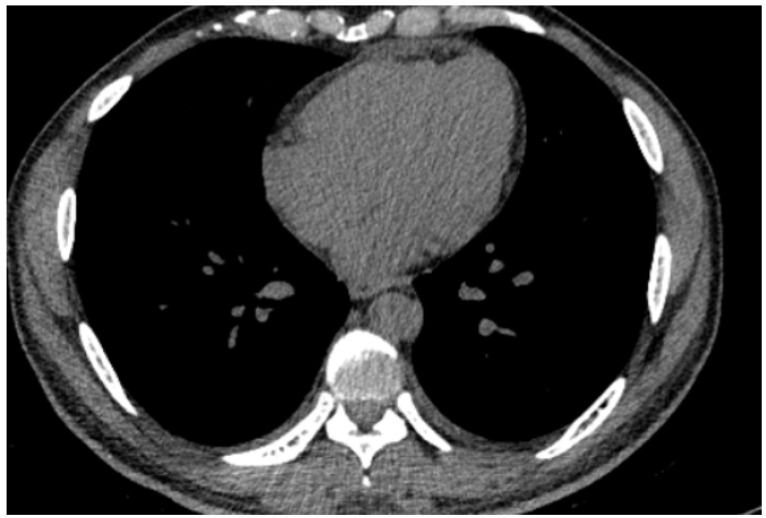
Cardiac CT scan findings. Slight thickening of the pericardium in the anterior and posterior regions, measuring up to 8 mm, with fluid density suggestive of mild pericardial effusion.

**Figure 4 jcm-14-00920-f004:**
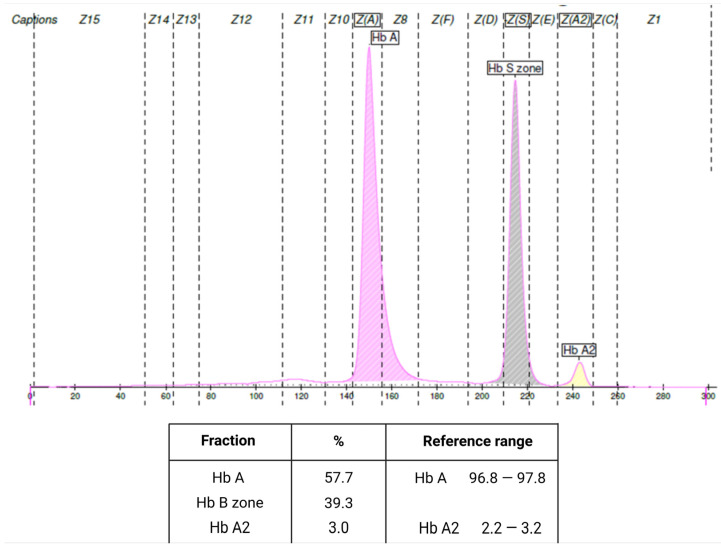
Hemoglobin electrophoresis.

**Figure 5 jcm-14-00920-f005:**
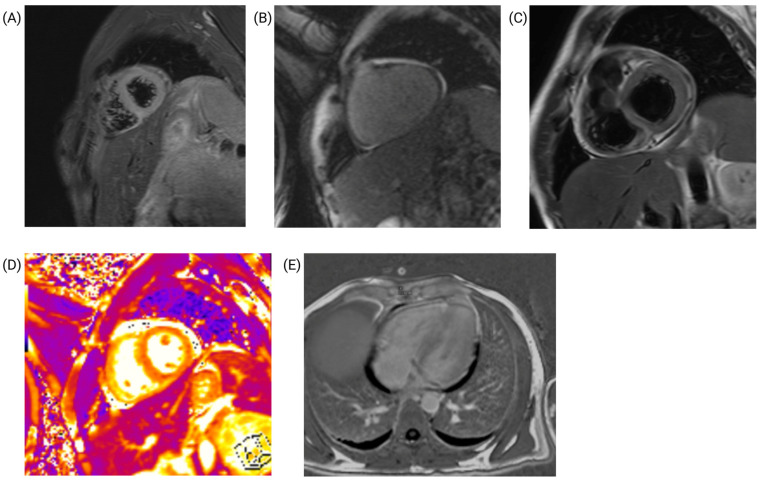
Cardiac magnetic resonance imaging findings. (**A**) T2-STIR (short axis) showing edema and inflammation in the cardiac apex. (**B**) Late gadolinium enhancement (sagittal view, short axis) showing pericardial thickening measuring approximately 5 mm. (**C**) T2-STIR (short axis) showing pericardial thickening. (**D**) T2 map (short axis) showing pericardial thickening measuring approximately 5 mm. (**E**) Late gadolinium enhancement (four-chamber, axial-view, long-axis) showing pericardial thickening and effusion.

**Table 1 jcm-14-00920-t001:** Clinical–hematological profile.

Test	Results	Reference Range
WBC	4.3 × 10^3^/μL	3.4–9.7 × 10^3^/μL
Neutrophil	3.17 × 10^3^/μL	2.2–4.8 × 10^3^/μL
Lymphocyte	0.93 × 10^3^/μL	1.1–3.2 × 10^3^/μL
Hemoglobin	12.5 g/dL	14.0–18.0 g/dL
Platelet	349,000/μL	130,000–400,000/μL
Troponin I	31.5 ng/mL	Positive: >25 ng/mL
Poikilocytosis	Positive	Not applicable
Double erythrocyte population	Positive	Not applicable
Antinuclear Ab (ANA)	1:1280	Positive: >1:80
Anti-citrullinated peptide Ab (ANTI CCP)	0.5 NTU	Positive: >11.0 NTU
Anti-RNP/Sm Ab	>200.0 IU/mL	Positive: >26 IU/mL
Creatine phosphokinase (CPK)	40.2 U/mL	Positive: >5.0 U/mL
Lactate dehydrogenase (LDH)	483 U/L	135–225 U/L
C3	123 mg/dL	90–180 mg/dL
C4	30.1 mg/dL	10–40 mg/dL

## Data Availability

Due to the nature of this case report, the data supporting the findings cannot be shared publicly in order to protect patient confidentiality. However, further details may be made available by the corresponding author upon reasonable request.

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
