# Peer review of "Myopericarditis and Pericardial Effusion as the Initial Presentation of Systemic Lupus Erythematosus in a Patient with Sickle Cell Trait: A Case Report"

_jcm, 2025, doi:10.3390/jcm14030920_

Round 1

Reviewer 1 Report

Comments and Suggestions for Authors

This is a rare but severe lupus case with sickle cell trait complicated by myocarditis and pericarditis. The case was successfully diagnosed and managed. However, I have several questions.

1.     In the title, the diagnosis of “myopericarditis” is used. However, in the article, most of the description focused on pericarditis. More details about myocarditis should be added.

2.     Complement levels such as C3, C4 should be reported.

3.     Did you perform pericardiocentesis to make further differential diagnosis? Such as infections or tumor? Especially when the patient was a middle-aged man.

4.     Line 120, “respiratory culture identified penicillium colonization, prompting treatment with itraconazole”. How about the treatment response and how long was the anti-infection duration?  What made the authors to decide to give the patient intravenous methylprednisolone pulses? The detailed dosage of methylprednisolone and IVIG should be added.

5.     Since there was a drop on hemoglobin, what’s your findings in blood smear? Did you see sickled RBC on it? Is there any correlation between sickle cell trait and myopericariditis?

6.     The literature review about the clinical characteristics of SLE complicated with sickle cell trait should be described in detail.

Author Response

Dear reviewers, we sincerely thank you for your efforts in searching for the best version of our manuscript. We have worked hard to respond to each of your comments and suggestions. Below is a point-by-point response to each comment, and we have attached the revised version of our manuscript that includes the changes in a different font color for easy identification. 

Reviewer 1

This is a rare but severe lupus case with sickle cell trait complicated by myocarditis and pericarditis. The case was successfully diagnosed and managed. However, I have several questions.

  1. In the title, the diagnosis of “myopericarditis” is used. However, in the article, most of the description focused on pericarditis. More details about myocarditis should be added.

Thank you for your comment. We understand the reviewer's point of view, in this context it is important to mention that certain clinical manifestations of acute myocarditis overlap with the manifestations of pericarditis, likewise, it is important to consider that both can and often do present concomitantly, i.e. as myopericarditis. We have added more information and expanded the discussion on the cardiac involvement of the patient by adding differential points of view to enrich the scientific literature.

  1. Complement levels such as C3, C4 should be reported.

Thank you for your comment. We have added the C3: 123 (90-180) and C4: 30.1 (10-40) levels in the case presentation.

  1. Did you perform pericardiocentesis to make further differential diagnosis? Such as infections or tumor? Especially when the patient was a middle-aged man.

Thank you for your comment. We did not perform a pericardiocentesis due to the minimal volume of pericardial fluid, as we felt that drainage or biopsy would pose a higher risk of complications for the patient. Instead, infections were ruled out by serial blood cultures, all of which yielded negative results. In addition, serological tests for HIV, hepatitis B and C, cytomegalovirus, herpes virus, rubella, toxoplasmosis and syphilis (VDRL) were performed and were negative. Cancer-focused studies were also performed. A CT scan with and without contrast revealed no evidence of masses or neoplasms, and tumor markers such as AFP, PSA, CA125, CA153 and CEA were within normal ranges, further ruling out neoplasms. This information has been clarified in the manuscript.

  1. Line 120, “respiratory culture identified penicillium colonization, prompting treatment with itraconazole”. How about the treatment response and how long was the anti-infection duration?  What made the authors to decide to give the patient intravenous methylprednisolone pulses? The detailed dosage of methylprednisolone and IVIG should be added.

Thank you for your comment. The antifungal treatment for the penicillium colonization was itraconazole at a dose of 200 mg twice daily for 1 week. This colonization was asymptomatic and was identified during routine evaluation prior to initiation of immunosuppressive therapy. Bronchoscopy with bronchoalveolar lavage was performed to detect tuberculosis, bacteria, and fungi. Real-time PCR for tuberculosis was negative, and cultures grew only Penicillium chrysogenum. On the recommendation of rheumatology specialists, after antifungal therapy, treatment for SLE was initiated. In addition, the patient received three intravenous pulses of methylprednisolone at a dose of 250 mg daily for three days, followed by a progressive reduction to oral prednisone. Mycophenolate mofetil was subsequently introduced to maintain disease control. In addition, the patient received two doses of intravenous immunoglobulin (IVIG) at a standard dose of 0.4 g/kg per dose. This treatment regimen produced an overall improvement in the patient's symptoms and clinical status. This information has been clarified in the manuscript.

  1. Since there was a drop on hemoglobin, what’s your findings in blood smear? Did you see sickled RBC on it? Is there any correlation between sickle cell trait and myopericariditis?

Thank you for your comment. The peripheral blood smear showed poikilocytosis and a dual red cell population, but no sickle red cells were seen. In addition, a metabisulfite test was performed, with negative result. A hemoglobin electrophoresis was performed due to the patient's family history, with results consistent with the diagnosis of the case.

Regarding the possible relationship between sickle cell disease and myopericaditis, within the spectrum of sickle cell manifestations, cardiac complications have been described, especially due to anemia and vasculopathy, with a high mortality rate of 20-32%, mainly associated with progressive heart failure. In addition, chronic anemia contributes to a high cardiac output state, which can lead to cardiac remodeling and eventually to left ventricular failure. Pulmonary hypertension caused by vasculopathy is another frequent complication. However, although these cardiac complications are well documented, no established direct association between sickle cell disease and myocarditis has been described. All this has been clarified in the manuscript.

  1. The literature review about the clinical characteristics of SLE complicated with sickle cell trait should be described in detail.

Thank you for your comment. This unique intersection of autoimmune and genetic conditions creates a complex clinical picture characterized by overlapping and exacerbating complications. In this context, we have expanded the discussion of the clinical features of SLE complicated with sickle cell trait to provide a more complete picture of our report and broaden the diagnostic picture of this condition. 

Reviewer 2 Report

Comments and Suggestions for Authors

The manuscript presents a rare clinical case of an Afro-Ecuadorian patient with systemic lupus erythematosus (SLE) and sickle cell trait (SCT), emphasizing the diagnostic challenges posed by the initial presentation of myopericarditis and pericardial effusion. Through a multimodal approach, which included cardiac MRI and immunosuppressive therapy, the diagnosis was made and the patient showed significant clinical improvement. Although the study highlights the rarity of this condition, several areas could benefit from some clarification.

COMMENTS:

  1. The general purpose of the study is stated, but it lacks precision regarding its novelty. A clearer definition of how this case uniquely contributes to the existing literature would increase its impact.
  2. The diagnostic challenges faced by the authors are mentioned but not explored in depth. A more detailed discussion of the specific factors that complicated the diagnosis, such as overlapping symptoms of SLE and SCT, would provide valuable insights into the complexity of the case.
  3. The rationale behind the chosen treatment regimen, particularly the use of mycophenolate mofetil, is not explicitly explained. Clarifying why this drug combination was chosen and how it aligns with current treatment guidelines would add depth to the discussion.
  4. While noting the patient's clinical improvement, the manuscript does not adequately address long-term prognosis or potential complications. The inclusion of a discussion of expected outcomes and challenges in managing similar cases would provide a more complete picture.
  5. The discussion mentions the rarity of SLE in men but does not elaborate on gender differences in presentation and management. Exploration of this topic would provide valuable insights.

Author Response

Dear reviewers, we sincerely thank you for your efforts in searching for the best version of our manuscript. We have worked hard to respond to each of your comments and suggestions. Below is a point-by-point response to each comment, and we have attached the revised version of our manuscript that includes the changes in a different font color for easy identification. 

Reviewer 2

The manuscript presents a rare clinical case of an Afro-Ecuadorian patient with systemic lupus erythematosus (SLE) and sickle cell trait (SCT), emphasizing the diagnostic challenges posed by the initial presentation of myopericarditis and pericardial effusion. Through a multimodal approach, which included cardiac MRI and immunosuppressive therapy, the diagnosis was made and the patient showed significant clinical improvement. Although the study highlights the rarity of this condition, several areas could benefit from some clarification.

 COMMENTS:

  1. The general purpose of the study is stated, but it lacks precision regarding its novelty. A clearer definition of how this case uniquely contributes to the existing literature would increase its impact.

Thank you for your insightful comment. We have addressed your concern by emphasizing the uniqueness and significance of our case report. Specifically, the clinical presentation, diagnostic challenges, and management of this case highlight its novelty, particularly due to the rare initial presentation of SLE perimyocarditis with concurrent sickle cell disease.

This report contributes to the existing literature by shedding light on rare conditions in specific population groups, exploring a potential correlation between SLE and sickle cell disease, and encouraging further observation of similar cases in high-risk regions and populations. To underscore its importance, we have expanded the manuscript to provide a more detailed discussion of these aspects, reinforcing the relevance and potential implications of our findings.

2. The diagnostic challenges faced by the authors are mentioned but not explored in depth. A more detailed discussion of the specific factors that complicated the diagnosis, such as overlapping symptoms of SLE and SCT, would provide valuable insights into the complexity of the case.

Thank you for your insightful comment. Diagnosing the overlap between SLE and SCT presents significant challenges due to the clinical and laboratory similarities between the two conditions. For example, hemolytic anemia may result from autoimmune hemolysis in SLE or hypoxia-induced sickling phenomena in SCT, while renal involvement, such as proteinuria and hematuria, could indicate lupus nephritis or SCT-associated nephropathy. Vascular complications, including thrombotic episodes, may arise from antiphospholipid syndrome in SLE or microvascular occlusion related to SCT. Furthermore, common symptoms like fatigue, joint pain, and Raynaud’s phenomenon can obscure the distinction between these pathologies, delaying diagnosis. Advanced diagnostic tools, such as autoantibody panels, hemoglobin electrophoresis, and renal biopsy, are often required to differentiate these overlapping conditions. We have expanded the discussion in the manuscript to provide greater depth and clarity on the specific diagnostic complexities involved in this case.

3. The rationale behind the chosen treatment regimen, particularly the use of mycophenolate mofetil, is not explicitly explained. Clarifying why this drug combination was chosen and how it aligns with current treatment guidelines would add depth to the discussion.

Thank you for your thoughtful comment. The selection of mycophenolate mofetil (MMF) as a key component of treatment was informed by its established efficacy in managing SLE and its alignment with current treatment guidelines, particularly for lupus nephritis. MMF is recommended as a first-line immunosuppressive agent due to its favorable efficacy-to-safety profile compared to alternatives like cyclophosphamide. Additionally, its availability in our region and its relatively lower risk of adverse effects, such as infections or hematologic toxicity, make it a suitable choice for patients with SCT, where minimizing such risks is critical. This regimen was specifically tailored to address overlapping symptoms of SLE and SCT, reduce inflammation, and prevent disease progression, while ensuring careful monitoring for potential complications and drug interactions. We have expanded on the rationale for the chosen treatment regimen, specifically the use of mycophenolate mofetil, to provide a more complete explanation within the manuscript

4. While noting the patient's clinical improvement, the manuscript does not adequately address long-term prognosis or potential complications. The inclusion of a discussion of expected outcomes and challenges in managing similar cases would provide a more complete picture.

Thank you for your thoughtful comment. The long-term prognosis in cases like this is shaped by the interplay between SLE and SCT, with increased risks of recurrent complications such as thrombotic events, progressive renal dysfunction, chronic anemia, and pulmonary hypertension. Effective management requires a personalized approach that balances immunosuppression to control SLE activity while minimizing the risks of SCT-related complications, including vaso-occlusive crises and infections. Additional challenges include preventing organ damage, particularly to the kidneys and cardiovascular system, as well as addressing the heightened risk of cerebrovascular events. Long-term care necessitates regular, multidisciplinary follow-up involving rheumatology, hematology, and nephrology to optimize outcomes and provide preventive care. We have expanded the manuscript to include a detailed analysis of the expected long-term prognosis, potential complications, and the need for individualized treatment strategies in cases of SLE complicated by SCT.

5. The discussion mentions the rarity of SLE in men but does not elaborate on gender differences in presentation and management. Exploration of this topic would provide valuable insights.

Thank you for your valuable suggestion. In this context, while SLE predominantly affects women, men with SLE often exhibit more severe disease manifestations, including higher rates of renal and cardiovascular involvement. These differences may be due to delayed diagnosis, as SLE is less frequently suspected in men, as well as hormonal and genetic factors that influence disease expression. Additionally, men may have an increased risk of treatment-related adverse effects, such as infections, associated with the use of immunosuppressive therapies. Recognizing these gender-based disparities is essential for improving diagnostic accuracy and tailoring treatment approaches. We have expanded the discussion to provide a detailed exploration of gender differences in the presentation and treatment of SLE, with a specific focus on male patients like the one in our case.

Round 2

Reviewer 1 Report

Comments and Suggestions for Authors

Thank you for your revision. I have no further questions.

Reviewer 2 Report

Comments and Suggestions for Authors

The authors responded thoroughly to my comments and revised the manuscript based on the comments.